# Evaluation of the Feasibility and Effectiveness of Placement of Fully Covered Self-Expandable Metallic Stents via Various Insertion Routes for Benign Biliary Strictures

**DOI:** 10.3390/jcm10112397

**Published:** 2021-05-28

**Authors:** Ko Tomishima, Shigeto Ishii, Toshio Fujisawa, Muneo Ikemura, Mako Ushio, Sho Takahashi, Wataru Yamagata, Yusuke Takasaki, Akinori Suzuki, Koichi Ito, Keiichi Haga, Kazushige Ochiai, Osamu Nomura, Hiroaki Saito, Tomoyoshi Shibuya, Akihito Nagahara, Hiroyuki Isayama

**Affiliations:** Department of Gastroenterology, Graduate School of Medicine, Juntendo University, 3-1-3 Hongo, Bunkyo-ku, Tokyo 113-0033, Japan; tomishim@juntendo.ac.jp (K.T.); sishii@juntendo.ac.jp (S.I.); t-fujisawa@juntendo.ac.jp (T.F.); m-ikemura@juntendo.ac.jp (M.I.); m-ushio@juntendo.ac.jp (M.U.); sho-takahashi@juntendo.ac.jp (S.T.); w.yamagata.mx@juntendo.ac.jp (W.Y.); ytakasa@juntendo.ac.jp (Y.T.); suzukia@juntendo.ac.jp (A.S.); kitoh@juntendo.ac.jp (K.I.); khaga@juntendo.ac.jp (K.H.); k.ochiai.qd@juntendo.ac.jp (K.O.); onomura@juntendo.ac.jp (O.N.); hiloaki@juntendo.ac.jp (H.S.); tomoyosi@juntendo.ac.jp (T.S.); nagahara@juntendo.ac.jp (A.N.)

**Keywords:** endoscopic treatment, benign biliary stricture, fully covered self-expandable metallic stents, metal stent, biliary stent

## Abstract

Background and aims: The goals of the management of benign biliary stricture (BBS) are to relieve symptoms and resolve short-/long-term stricture. We performed fully covered self-expandable metallic stent (hereafter, FCSEMS) placement for BBS using various methods and investigated the treatment outcomes and adverse events (AEs). Methods: We retrospectively studied patients who underwent FCSEMS placement for refractory BBS through various approaches between January 2017 and February 2020. FCSEMS were placed for 6 months, and an additional FCSEMS was placed if the stricture had not improved. Technical success rate, stricture resolution rate, and AE were measured. Results: A total of 26 patients with BBSs that were difficult to manage with plastic stents were included. The mean overall follow-up period was 43.3 ± 30.7 months. The cause of stricture was postoperative (46%), inflammatory (31%), and chronic pancreatitis (23%). There were four insertion methods: endoscopic with duodenoscopy, with enteroscopy, EUS-guided transmural, and percutaneous transhepatic. The technical success rate was 100%, without any AE. Stricture resolution was obtained in 19 (83%) of 23 cases, except for three cases of death due to other causes. Stent migration and cholangitis occurred in 23% and 6.3%, respectively. Stent fracture occurred in two cases in which FCSEMSs were placed for more than 6 months (7.2 and 10.3 months). Conclusion: FCSEMS placement for refractory BBS via various insertion routes was feasible and effective. FCSEMSs should be exchanged every 6 months until stricture resolution because of stent durability. Further prospective study for confirmation is required, particularly regarding EUS-guided FCSEMS placement.

## 1. Introduction

There are various causes of benign biliary stricture (BBS), and their characteristics and the clinical course vary according to the etiology. The exclusion of pathological malignancy is essential for diagnosis, and minimally invasive endoscopic treatment is the first choice for treatment [1,2,3]. The goal of treatment for BBS is to obtain long-term bile duct patency without a stent. Fully covered self-expandable metallic stents (hereafter, FCSEMS) have been used safely, and the stricture resolution rate is better than with PSs (85.4% vs. 92.6%, *p* < 0.001) [4]. In addition, the number of ERCP sessions for stricture release is also significantly fewer, which may reduce the patient’s burden (3.13 vs. 2.21, *p* < 0.0001) [4].

BBS may be caused by inflammatory changes such as chronic pancreatitis (CP) or choledocholithiasis or by surgical procedures such as after liver transplantation or cholecystectomy. Other causes include trauma and infection. Strictures are common in the distal bile duct in chronic pancreatitis and in the vicinity of the hilum after cholecystectomy. Approaches to strictures include trans-papillary in some cases, such as in chronic pancreatitis, and via bilioenteric anastomotic strictures in postoperative cases. In postoperative cases, the location and approach of the stricture vary depending on the surgical reconstruction procedures. Similarly, it is important whether the anastomotic site has 1 or 2 holes. Repeated surgery, including hepatectomy, may not reveal the anastomotic site. In such cases, balloon-assisted enteroscopy (BAE) may not succeed because of strong adhesion to the intestine. A meta-analysis reported that 61.7% of BAE patients achieve successful biliary intervention [5]. EUS-guided transmural drainage and percutaneous transhepatic biliary drainage (PTBD) are adopted when duodenoscopy/BAE fails. EUS-guided transmural drainage is less painful than PTBD, but it can be difficult to approach the right intrahepatic bile duct. On the other hand, PTBD is associated with discomfort when dilating the puncture route. FCSEMS insertion using duodenoscopy and PTBD is technically not so difficult. In case of BAE, once reached to the stricture and passing the stricture, FCSEMS can be inserted in most cases. However, there are few reports of the insertion route via EUS-guided transmural drainage. It is important to recognize etiology of stricture and which approach route toward stricture is best for patient from the viewpoint of patient’s burden. In this study, we performed FCSEMS placement for BBS at our hospital using various methods: duodenoscopy, BAE, EUS-guided transmural drainage, and PTBD. We investigated the treatment outcomes and complications.

## 2. Materials and Methods

### 2.1. Study Design

This study was conducted as a single-center retrospective analysis and was approved by our institutional review board (ethical code 20-306). Informed consent to participate in the study was obtained in the form of an opt-out on the website of Juntendo University. We reviewed the charts and database of endoscopy and radiological procedures. All authors had access to the study data and approved the final manuscript.

### 2.2. Methods

The inclusion criteria of this study were as follows: refractory bile duct strictures that could not be resolved by PS placement between January 2017 and February 2020 and the patient was followed up for more than 6 months at our hospital. All patients were indicated to insert PS to control cholangitis. The exclusion criteria were malignant disease diagnosed by pathological examination, cholangioscopy, or clinical course and patients who could not consent to the procedure. The placement of FCSEMS was considered in the order of duodenoscopy, BAE, EUS-guided transmural drainage, and PTBD. First, a FCSEMS could be inserted through the papilla or the anastomotic site with a duodenoscopy. BAE was performed first for patients with altered gastrointestinal anatomy. If BAE failed due to adhesions, EUS-guided transmural drainage was considered. PTBD was performed for cases in which both EUS-guided transmural drainage and BAE were impossible. The FCSEMS used in this study was BONASTENT M-Intraductal (Standard Sci-Tech Inc., Seoul, Korea), 10 mm in diameter. The stent is made of nitinol; the diameter at both ends is 10 mm, and the central diameter is 8 mm. The delivery system is 8 Fr, and both 0.025- and 0.035-inch guide wires are available. The FCSEMS can be removed by pulling the lasso at the proximal, duodenal side of the stent. After approximately 6 months of implantation, all FCSEMS s were removed, and the stricture was evaluated. If the stricture remained, the FCSEMS was reinserted and placed for 6 additional months.

### 2.3. Endoscopic and Percutaneous Procedure

Most patients underwent endoscopic procedures using a standard duodenoscope (TJF 260, JF 260; Olympus Optical Co., Ltd., Tokyo, Japan or ED580T; FUJIFILM, Tokyo, Japan), double-balloon endoscopy (EI-580BT; FUJIFILM, Tokyo, Japan), or EUS scope (EG580UT; FUJIFILM, Tokyo, Japan). For endoscopic procedures, local anesthesia of the pharynx was performed with 8% lidocaine; patients were sedated with an intravenous injection of pentazocine (15 mg) and midazolam (2–10 mg) before scope insertion. All patients received endoscopic procedures in the prone position, and the sedative drugs were monitored during the procedure. Small endoscopic sphincterotomy (EST) following balloon dilation if needed was performed for patients from transpapillary insertion after exclusion of malignancy. Following insertion of the guidewire via the stricture (Figure 1a), the FCSEMS was inserted (Figure 1b), and the central X-marked portion of the stent was positioned at the center of the stricture (Figure 1c). In cases of BAE, balloon dilation was adopted when the stent delivery system could not pass through the stricture. The additional PS was placed in the contralateral hepatic duct to prevent segmental cholangitis, which was caused by obstruction of the biliary branch in cases of hilar stricture. Six months after initial stenting, the stent was removed through the working channel of a therapeutic scope by grasping the lasso using forceps [6]. If the stricture was not resolved, a new FCSEMS was placed for an additional 6 months.

### 2.4. Outcome Measurement and Statistical Analyses

Technical success was defined as precise placement of the FCSEMS in the center of the stricture. It was also defined as good drainage to eliminate contrast medium and bile juice along the FCSEMS (Figure 1d). Clinical success was defined as a clinical course without cholangitis after stent removal. Stent migration was defined as complete stent displacement above or below the stricture, with or without symptoms. However, the time points of re-intervention were defined as the points at which symptoms associated with occlusion or migration were observed, that is, time to recurrent biliary obstruction [7]. An early adverse event (within 30 days) was defined as any AE related to stent insertion, such as cholangitis, cholecystitis, pancreatitis, bleeding, or perforation. All adverse events were classified and graded according to the TOKYO criteria 2014 [7].

Statistical analyses were performed using BellCurve for Excel statistical software (Microsoft Inc., Redmond, WA, USA). Data are presented as mean and standard deviation or median with range. The data were analyzed using Fisher’s exact probability test and the Mann–Whitney U test. A two-sided *p*-value < 0.05 was considered significant.

## 3. Results

### 3.1. Patient Characteristics and Etiology of BBSs

A total of 26 patients were included as cases with refractory BBS that involved previously placed PSs. The male:female ratio was 19:7, and the mean age was 66.3 years. The mean overall follow-up period was 43.3 ± 30.7 months, the period of previous PS placement was 22.2 months, and the observation period after FCSEMS placement was 21.9 months (Table 1). The cause of the stricture was postoperative in 12 patients (46%), inflammation in 8 patients (31%), and CP in 6 patients (23%) (Table 2). Postoperative strictures included 4 cases of hepaticojejunostomy anastomosis stricture, 4 cases of perihilar stricture after cholecystectomy, 2 cases of choledochoduodenostomy anastomosis stricture, and 2 cases of perihilar stricture after hemihepatectomy. Inflammatory strictures occurred in 2 cases of hepatolithiasis, 2 cases of iatrogenicity, and single cases of IgG4-related sclerosing cholangitis (IgG4-SC), Caroli’s disease, walled-off necrosis after pancreatitis, and unknown etiology. Duodenoscopy was the FCSEMS insertion route in 20 cases (18 cases of transpapillary, 2 cases of transanastomotic), BAE in 4 cases (only transanastomotic), and EUS-guided transmural drainage, PTBD in one case (Table 3).

### 3.2. Procedure

The number of endoscopic sessions was reduced from once every 2.3 months with previous PS to once every 5.3 months before and after FCSEMS. The average number of placed PSs before FCSEMS was an average of 1.7 stents. All patients could insert a FCSEMS with a technical success rate of 100%. The stents were replaced every six months to evaluate stricture. Stent removal was performed if the stricture was improved. The excluded 3 cases could not be followed up because of death from other causes. It was possible to remove the FCSEMS in 19 cases (73%). Six months after FCSEMS implantation, stricture resolution was achieved in 17 patients, including 8 with postoperative stricture (47%), 5 with inflammatory stricture (29%), and 4 with CP (24%). Removal was possible at 12 months and 18 months in two patients with postoperative stricture. The mean time to removal was 7.3 months, and the time to recurrent biliary obstruction (TRBO) after removal was 13.6 months. There were four patients in whom the stent could not be removed, and two patients had re-stricture after removal. Patients with re-stricture had CP and postoperative left bile duct stenosis, and the time to recurrent obstruction was 1.2 months and 17 months, respectively (Table 4). The cases in which removal was not possible tended to have longer strictures than those in which removal was possible (28.3 mm vs. 16.3 mm, *p* = 0.09).

One patient who failed BAE was changed to EUS-guided transmural drainage. First, one PS was placed through EUS-guided hepaticogastrostomy (EUS-HGS), and the fistula was dilated with two PSs 3 months later. Eleven months later, an FCSEMS was placed from the EUS-HGS route to the anastomotic side, and a PS bridging from the left to the right hepatic duct was placed in the stomach. The lasso of the FCSEMS was put in the stomach through the HGS route, and the gastric wall and the lasso were fixed with the clip to prevent stent migration (Figure 2). PTBD was performed for one patient for whom BAE and EUS-guided transmural drainage was not possible. The first PTBD was implanted with 8 Fr, and 2 months later, a percutaneous FCSEMS was placed on the anastomotic site. The lasso was fixed to the skin in this case.

As for the outcome of each placement method, all the technical success rates were 100% and a 100% clinical success was obtained in the EUS-guided and BAE methods. Percutaneous method still inserted FCSEMS because of the short follow up periods (6.4 months). On the other hand, 12 cases of twenty (60%) resolve the stricture using the duodenoscopy method.

### 3.3. Adverse Events

For the analyses of AEs, 48 sessions of 26 patients were considered (Table 5). Stent dislocation occurred in 11 sessions (23%), and fall-off and inside migration occurred in 5 and 6 sessions, respectively. All cases were asymptomatic. In five cases with migration into the bile duct, the stents were easily removed by grasping the lasso with a forceps. In addition, stent injury occurred in 2 patients, and the time to replacement was 7.2 months and 10.3 months, respectively. Post-procedural cholangitis as an AEs occurred in 3 patients (6.3%), all of whom had cholangitis due to FCSEMS obstruction of the intrahepatic bile duct. Severity grading of all cholangitis was moderate, and re-intervention was performed within several days. Hyperamylasemia, which improved spontaneously, was observed 2 days after the FCSEMS placement.

## 4. Discussion

The placement of FCSEMSs via various insertion routes was feasible and effective to resolve benign biliary strictures that could not be controlled by PSs. Seventeen patients (89%) improved after the first 6 months, suggesting that use of an FCSEMS, rather than a PS, may be useful during hospitalizations. On the other hand, the 4 patients for whom FCSEMS s could not be removed included 2 with CP and 2 with postoperative stricture. Strictures due to CP are more difficult to treat endoscopically compared to other types of BBSs, particularly in patients with calcific CP [8]. Two patients in this study had calcifications in the head of the pancreas. In addition, considering continuous drinking and the length of the stricture, current drinking and long stricture (more than 30 mm) tended to be associated with poor patency (*p* = 0.07). Lakhtakia et al. revealed that the stricture length was also a significant predictor of stricture recurrence in multivariate analyses (HR, 1.2; 95% CI, 1.0–1.4; *p* = 0.022) [9]. In one of two patients with postoperative strictures, B2 intrahepatic bile duct stricture was not improved. The other patient received an FCSEMS through the PTBD route for hepaticojejunostomy anastomosis stricture after pancreatoduodenectomy. A prolapsed coil for vascular embolization caused the stricture in the bile duct in this case. Five patients with inflammatory stricture had patency after semiannual insertion of FCSEMS s. In summary, difficult management of biliary stricture was as follows: CP with continuous drinking and stricture length over 30 mm, and postoperative cases with strictures of the intrahepatic bile duct and retained foreign substances. There was previously little evidence on FCSEMS treatment for bilioenteric anastomotic stricture in patients with surgically altered anatomy [10]. Recently, for patients with perihilar BBSs, such as stricture following liver transplantation and hepaticojejunostomy anastomotic stricture, a temporary placement of FCSEMS s for refractory BBSs was found to be safe and effective [11]. All 4 patients showed clinical improvement, and no major complications were observed in this study. The stent dislocation rate was almost the same compared with previous studies [12,13]. Stent dislocation tends to occur in patients with resolve the stricture cases. All patients with dislocated stents have nothing with symptom and their strictures were all resolved in this study.

Although the route of FCSEMS insertion from the PTBD and EUS-AI involved only one patient each, it was technically feasible to place the FCSEMS. There are three concerns associated with placement of a covered metal stent from the PTBD and EUS-guided transmural drainage route. First, the placement of the FCSEMS at the choledochojejunostomy anastomotic stricture may cause cholangitis in the contralateral bile duct. Second, the timing of FCSEMS insertion should be considered. The third involves the management of the lasso. In patients with PTBD, stent insertion was performed after expanding to 10 Fr over a period of 2 months, and the stent was successfully implanted without any complications. For EUS-guided transmural drainage, two PSs were inserted approximately 3 months after the first puncture. And FCSEMSs could be successfully placed through the matured route. Two-step EUS-guided transmural drainage after puncture route maturation, resulting in the prevention of bile leakage, can be considered safer than 1-step EUS-guided transmural drainage [14]. This method appears to be a feasible and safe alternative procedure after BAE failure [14]. This type of FCSEMS may be adopted with a lasso for removal. In EUS-guided transmural drainage, the lasso is fixed to the gastric wall with clips, and in PTBD, it is fixed to the skin to prevent dislocation. If the lasso is cut and the FCSEMS migrates to the intestinal side, removal is possible with percutaneous insertion of a cholangioscope [15]. In PTBD with PS, prospective data showed that percutaneous catheter placement for 6 to 8 months in extrahepatic, biliary strictures, with progressive catheter insertion upsizing to 18F to 20F, was effective and achieved stricture resolution in 86.4% of patients with BBSs [16]. However, considering the pain associated with dilation and the decrease in activities of daily living (ADL) in patients, long-term PTBD placement should be avoided.

Stent damage associated with long-term deployment is a problem with FCSEMSs. In this study, there were 2 cases in which the stent was fractured with stent removal. Both were indwelling for more than 6 months (7.2 and 10.3 months), and it is recommended to replace the stents within 6 months. The appropriate indwelling time of an FCSEMS is still unclear, but the period has typically ranged from 3 to 6 months [4,17,18,19,20]. All three patients with post-FCSEMS cholangitis had obstruction of the intrahepatic bile duct branch. The recommendation to prevent segmental cholangitis is the placement of an additional PS in the opposite hepatic duct after deployment of the FCSEMS across the liver hilum [21,22]. Recurrence of the stricture appeared frequently in patients with CP. In patients with CP, 5 years after placement of a single FCSEMS intended for 10 to 12 months indwell, more than 60% remained asymptomatic and stent-free with an acceptable safety profile [9]. A longer indwelling period and the absence of stent migration might be important factors for stricture resolution [23]. With regard to CP, there is no difficulty with the re-intervention because of the transpapillary route. However, re-puncture for re-stricture in patients with PTBD and EUS-guided transmural drainage is difficult. As for each insert method, we could technically insert FCSEMS in all methods. From the viewpoint of patients’ burden, if possible, we tried EUS-BD or BAE over the PTBD method. The reason why clinical success of duodenoscopy was low, although enough follow up periods, is because refractory stricture of CP. Therefore, further investigation is needed about stricture resolution rate and long-term outcome for each method.

The average cost per patient for once hospitalization to exchange the stent is 819,200 yen for FCSEMS and 626,870 yen for multiple PSs. FCSEMS and multiple PSs were exchanged every 6 months and 3 months, respectively. Thus, total cost effectiveness of FCSEMS is better than PSs. We acknowledge several limitations of this study. This was a single-arm study without a control group of other modalities, so there may be a treatment selection bias. We included various etiologies of BBS; however, the sample size was not large enough to investigate the effectiveness of FCSEMSs for each etiology.

In conclusion, FCSEMS placement for refractory benign biliary stricture via various insertion routes (duodenoscopy, enteroscopy, EUS-guided, and percutaneous) was feasible and effective. FCSEMSs should be exchanged every 6 months until stricture resolution because of stent durability. Further prospective study for confirmation is required, particularly for EUS-guided FCSEMS placement.

## Figures and Tables

**Figure 1 jcm-10-02397-f001:**
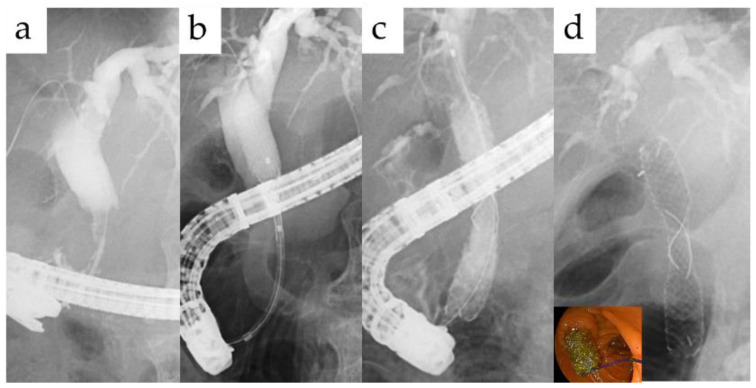
Stent placement method. Following insertion of the guidewire via the stricture (**a**), the FCSEMS was inserted (**b**), and the central X-marked portion of the stent was positioned at the center of the stricture (**c**). Confirm contrast medium to drainage along the FCSEMS (**d**).

**Figure 2 jcm-10-02397-f002:**
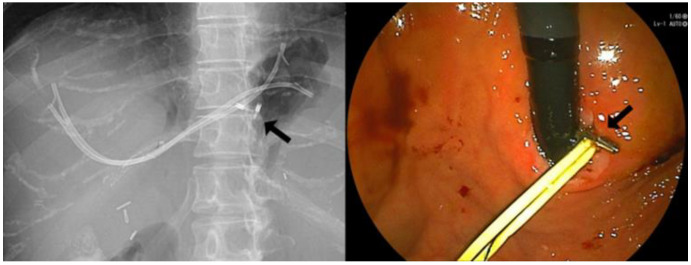
FCSEMS was placed from the EUS-HGS route to the anastomotic side, and a PS bridging from the **left** to the **right** hepatic duct was placed in the stomach. The Lasso of FCSEMS was fixed with the clip to the gastric wall (arrow).

**Table 1 jcm-10-02397-t001:** Patients’ characteristics.

Cases	26
Male/Female	19/7
Age, mean ± SD	66.3 ± 14.3
Mean overall follow-up period, months	43.3 ± 30.7
Mean period of plastic stent placement, month	22.2 ± 23.5
Mean period after SEMS insertion, month	21.9 ± 13.2

**Table 2 jcm-10-02397-t002:** Etiology of biliary stricture.

Etiology	Number (%)
Postoperative	12 (46)
Hepaticojejunostomy anastomosis stricture	4
Perihilar stricture after cholecystectomy	4
Choledochoduodenostomy anastomosis stricture	2
Perihilar stricture after hemihepatectomy	2
Inflammatory	8 (31)
Hepatolithiasis	2
Iatrogenicity	2
IgG4-related sclerosing cholangitis	1
Caroli’s disease	1
Walled-off necrosis after pancreatitis	1
Unknown	1
Chronic pancreatitis	6 (23)

**Table 3 jcm-10-02397-t003:** Fully covered self-expandable metallic stent (FCSEMS) insertion method.

FCSEMS Insertion Method		N
Duodenoscopy		20
	Trans-papillary	18
	Trans-anastomotic	2
BAE	Trans-anastomotic	4
Percutaneous		1
EUS-guided		1

BAE; balloon-assisted enteroscopy, EUS: endoscopic ultrasound.

**Table 4 jcm-10-02397-t004:** Details of the procedure.

The Number of Endoscopic Sessions	
Mean PS exchange period, month	2.3
Mean FCSEMS exchange period, month	5.3
Withdrawn by death due to other causes	3
Remove the FCSEMS, (%)	19 (73)
Mean time to removal, month	7.3 ± 4.9
Mean stent free period after FCSEMS removal	13.6 ± 9.9
Re-intervention for RBO after FCSEMS removal, (%)	2 (11)
TRBO after FCSEMS removal, m	1.2/17

FCSEMS: Fully covered self-expandable metallic stent, RBO: recurrent biliary obstruction, TRBO: Time to recurrent biliary obstruction.

**Table 5 jcm-10-02397-t005:** Adverse events.

Total Number of Procedures	48	
	Within 30 Days	31 Days or Later
Dislocation, (%)	0	11 (23)
Disappeared	0	5
Migration	0	6
Broken of the FCSEMS, (%)	0	2 (4.2)
Placement duration until FCSEMS broken, m	N/A	7.2/10.3
Cholangitis, (%)	3 (6.3)	0
Hyperamylasemia, (%)	1 (2.1)	0

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
