# Peer review of "Evaluation of the Feasibility and Effectiveness of Placement of Fully Covered Self-Expandable Metallic Stents via Various Insertion Routes for Benign Biliary Strictures"

_jcm, 2021, doi:10.3390/jcm10112397_

Round 1

Reviewer 1 Report

This is a paper of the feasibility and effectiveness of placement of full covered self-expandable metallic stents for benign biliary strictures (BBS). The effectiveness of self-expandable metallic stents for BBS was reported, but there was not enough evidence yet.

Minor points

  1. Author should discuss about the cost benefits of self-expandable metallic stent compared from plastic stents.
  2. The rate of stent dislocation was higher than that of previous reports. Author should discuss the reason.
  3. Author should check the calibration such as spaces.

Author Response

May 17, 2021

Journal of Clinical Medicine

Dear Editor; Rainie Zhang and Reviewers

Thank you very much for reviewing our manuscript and offering valuable advice. We have addressed your comments with point-by-point responses, and revised the manuscript accordingly. Please find the revised version of the manuscript entitled “jcm-1212505. Evaluation of the feasibility and effectiveness of placement of fully covered self-expandable metallic stents via various insertion routes for benign biliary strictures” with tables and figures to be considered for publication in Journal of Clinical Medicine.

Please contact me if there are further questions regarding this revised manuscript. We appreciate if decision of acceptance on this manuscript would be transferred by e-mail. Thank you for your consideration. We are looking forward to hearing from you.

Sincerely,

Ko Tomishima, M.D.

Hiroyuki Isayama, M.D., Ph.D.

Department of Gastroenterology,

Juntendo University, School of Medicine,

2-1-1, Hongo, Bunkyo-ku, Tokyo,

113-8421, Japan

Phone; +81-3-5802-1060

Fax; +81-3-3813-8862

E-mail; [email protected]

Manuscript ID ;jcm-1212505

Reviewer 1

Minor points

  1. Author should discuss about the cost benefits of self-expandable metallic stent compared from plastic stents.

→Thank you for reviewers suggestion, we corrected it as follow; “The average cost per patient for once hospitalization to exchange the stent is 819,200 yen for FCSEMS and 626,870 yen for multiple PSs. FCSEMS and multiple PSs were exchanged every 6 months and 3 months, respectively. Thus, total cost effectiveness of FCSEMS is better than PSs.” (P8, Line 282-285)

  1. The rate of stent dislocation was higher than that of previous reports. Author should discuss the reason.

→Thank you for valuable advice. Some papers reported almost same rate of stent migration (31-46.7 %) (Daisy Walter et al. 2015 GASTROINTESTINAL ENDOSCOPY and Tarantino et al. 2012 Endoscopy). So, we added as follow; “Stent dislocation rate is almost the same compared with previous studies12,13. Stent dislocation tends to occur in patients with resolve the stricture cases. All patients with dislocated stents have nothing with symptom and their strictures were all resolved in this study.” (P7, Line 236-239)

  1. Author should check the calibration such as spaces.

→Thank you for reviewers suggestion. We corrected.

Reviewer 2 Report

Recommendation: Major revisions

General Comment

The authors reported the biliary stent placement via various routes for benign strictures. The manuscript category and readability is okay. Before final acceptance, comments must be addressed.

Specific Comments

Introduction

  1. It should be focused on four different stent placement methods. The originality of this paper is a technical feasibility and safety of the placement techniques and its selection criteria. Clinical outcomes of FCSEMS for benign biliary strictures are well known.

Materials and Methods

  1. How to choice of stent placement method? Selection criteria should be described in Materials and Methods section.
  2. Can you compare the clinical and technical outcomes between the four placement methods?
  3. Temporary stent placement usually used for benign non-vascular strictures. Are there any protocols for indwell time of stent placement? When you removed the FCSEMS, are there any procedure related complications?

Results

  1. Please add representative figures of stent placement methods.
  2. Again, it seems necessary to describe the technical findings or the results according to the placement methods.
  3. It is necessary to describe the advantages and disadvantages of each method, and to provide evidence on which method is useful in some cases.

Author Response

May 17, 2021

Journal of Clinical Medicine

Dear Editor; Rainie Zhang and Reviewers

Thank you very much for reviewing our manuscript and offering valuable advice. We have addressed your comments with point-by-point responses, and revised the manuscript accordingly. Please find the revised version of the manuscript entitled “jcm-1212505. Evaluation of the feasibility and effectiveness of placement of fully covered self-expandable metallic stents via various insertion routes for benign biliary strictures” with tables and figures to be considered for publication in Journal of Clinical Medicine.

Please contact me if there are further questions regarding this revised manuscript. We appreciate if decision of acceptance on this manuscript would be transferred by e-mail. Thank you for your consideration. We are looking forward to hearing from you.

Sincerely,

Ko Tomishima, M.D.

Hiroyuki Isayama, M.D., Ph.D.

Department of Gastroenterology,

Juntendo University, School of Medicine,

2-1-1, Hongo, Bunkyo-ku, Tokyo,

113-8421, Japan

Phone; +81-3-5802-1060

Fax; +81-3-3813-8862

E-mail; [email protected]

Reviewer 2

Specific Comments

Introduction

  1. It should be focused on four different stent placement methods. The originality of this paper is a technical feasibility and safety of the placement techniques and its selection criteria. Clinical outcomes of FCSEMS for benign biliary strictures are well known.

→Thank you for valuable advice. We summarized the first of half of “Introduction” and corrected as follow; “FCSEMS insertion using duodenoscopy and PTBD is technically not so difficult. In case of BAE, once reached to the stricture and passing the stricture, FCSEMS can be inserted in most cases. But there are few reports of the insertion route via EUS-guided transmural drainage. It is important to recognize etiology of stricture and which approach route toward stricture is best for patient from the viewpoint of patient’s burden.” (P2, Line 59-64).

Materials and Methods

  1. How to choose of stent placement method? Selection criteria should be described in Materials and Methods section.

→We corrected it as follow; “The placement of FCSEMS was considered in the order of duodenoscopy, BAE, EUS-guided transmural drainage, and PTBD. First, a FCSEMS could be inserted through the papilla or the anastomotic site with a duodenoscopy. BAE was performed first for patients with altered gastrointestinal anatomy. If BAE did not succeed due to adhesions, EUS-guided transmural drainage was considered. PTBD was performed for cases in which both EUS-guided transmural drainage and BAE were impossible.” (P2 Line 80-86)

  1. Can you compare the clinical and technical outcomes between the four placement methods?

→We corrected it as follow; “As for outcome of each placement method, all technical success rate was 100% and clinical success was obtained 100% in EUS-guided and BAE methods. Percutaneous method is still inserted FCSEMS because of short follow up periods (6.4 months). On the other hand, 12 cases of twenty (60%) was resolve the stricture in duodenoscopy method.” (P6, Line 193-196)

  1. Temporary stent placement usually used for benign non-vascular strictures. Are there any protocols for indwell time of stent placement? When you removed the FCSEMS, are there any procedure related complications?

→There were 2 cases in which the stent was fractured with stent removal. Both were indwelling for more than 6 months (7.2 and 10.3 months), and it is recommended to replace the stents within 6 months. (P7 Line 263-265).

Results

  1. Please add representative figures of stent placement methods.

→Thank you for valuable advice. We added Figure 1.

  1. Again, it seems necessary to describe the technical findings or the results according to the placement methods.

→We corrected it as follow; “As for outcome of each placement method, all technical success rate was 100% and clinical success was obtained 100% in EUS-guided and BAE methods. Percutaneous method is still inserted FCSEMS because of short follow up periods (6.4 months). On the other hand, 12 cases of twenty (60%) was resolve the stricture in duodenoscopy method.” (P6, Line 193-196)

  1. It is necessary to describe the advantages and disadvantages of each method, and to provide evidence on which method is useful in some cases.

→We corrected it as follow; “As for each insert method, we could technically insert FCSEMS at all method. From the viewpoint of patient’s burden, if possible, we would try EUS-BD or BAE better than PTBD. The reason why clinical success of duodenoscopy was low, although enough follow up periods, is because refractory stricture of CP. Therefore, further investigation is needed about stricture resolution rate and long-term outcome for each method.” (P8, Line 276-281)

Round 2

Reviewer 2 Report

The quality of the revised manuscript was significantly improved.